# Foliar Application of Amino Acids and Nutrients as a Tool to Mitigate Water Stress and Stabilize Sugarcane Yield and Bioenergy Generation

**DOI:** 10.3390/plants13030461

**Published:** 2024-02-05

**Authors:** Lucas Moraes Jacomassi, Marcela Pacola, Letusa Momesso, Josiane Viveiros, Osvaldo Araújo Júnior, Gabriela Ferraz de Siqueira, Murilo de Campos, Carlos Alexandre Costa Crusciol

**Affiliations:** 1Department of Crop Science, College of Agricultural Science, São Paulo State University (UNESP), Botucatu 18610-034, SP, Brazil; lucas.jacomassi@unesp.br (L.M.J.); marcela.pacola@unesp.br (M.P.); josiane.v.oliveira@unesp.br (J.V.); osvaldo.araujo@unesp.br (O.A.J.); gaferrazsiq@gmail.com (G.F.d.S.); murilodecampos83@gmail.com (M.d.C.); 2Department of Agriculutre, School of Agriculture, Federal University of Goiás (UFG), Goiânia 74690-900, GO, Brazil; letusa.momesso@ufg.br

**Keywords:** *Saccharum* spp., oxidative stress, antioxidant enzymes, water deficit

## Abstract

Extended periods of water stress negatively affect sugarcane crop production. The foliar application of supplements containing specific nutrients and/or organic molecules such as amino acids can improve sugarcane metabolism, stalk and sugar yields, and the quality of the extracted juice. The present study assessed the effectiveness of the foliar application of an abiotic stress protection complement (ASPC) composed of 18 amino acids and 5 macronutrients. The experiments were carried out in the field with two treatments and twelve replicates. The two treatments were no application of ASPC (control) and foliar application of ASPC. The foliar application of ASPC increased the activity of antioxidant enzymes. The Trolox-equivalent antioxidant capacity (DPPH) was higher in ASPC-treated plants than in control plants, reflecting higher antioxidant enzyme activity and lower malondialdehyde (MDA) levels. The level of H_2_O_2_ was 11.27 nM g^−1^ protein in plants treated with ASPC but 23.71 nM g^−1^ protein in control plants. Moreover, the application of ASPC increased stalk yield and sucrose accumulation, thus increasing the quality of the raw material. By positively stabilizing the cellular redox balance in sugarcane plants, ASPC application also increased energy generation. Therefore, applying ASPC is an effective strategy for relieving water stress while improving crop productivity.

## 1. Introduction

The current search for renewable energy places Brazil as one of the main global players in the production of clean and sustainable energy due to its position as the world’s largest producer of sugarcane (*Saccharum* spp.) and ethanol derived from it as an alternative to fossil fuels [1,2,3]. Furthermore, the country also stands out in the production of sugar as a commodity and the cogeneration of electrical energy from burning sugar cane biomass. Consequently, the sugar-energy sector is one of the most important segments of Brazilian agribusiness, which makes the importance of this crop for the country’s economy irrefutable [4,5]. However, the seasonality of the climate, especially the drought regime, is a severe counterpoint to agricultural productivity [6,7]. This issue is intensified by the expansion of sugarcane in non-traditional areas of cultivation with greater vulnerability to atypical patterns of water deficit, such as the central region of Brazil. These areas have a very poor rainfall regime, with the rainy season being poorly distributed and water availability limited, especially between April and September, or even highly variable depending on the time of year [8,9,10]. Sugarcane needs an average precipitation of 1800 to 2200 mm per crop cycle, thus, as seasonal droughts are common in many regions of Brazil, sugarcane production and longevity are limited due to water stress on the plants [11,12].

Water stress results in changes in plant metabolism, morphology, and physiology [1,13,14]. Under water stress conditions, sugarcane and other plants activate stomatal closure to reduce plant transpiration, resulting in lower CO_2_ fixation and changes in photosynthetic activity that increase ROS production beyond the antioxidant capacity of cells [15]. ROS cause oxidative damage to proteins, DNA, RNA, and lipids and include superoxide (O_2_^−^), hydrogen peroxide (H_2_O_2_), hydroxyl radical (OH^−^), and singlet oxygen (^1^O_2_) [16]. At low concentrations in the absence of stress, ROS act as signaling molecules. However, at high concentrations, ROS are highly oxidizing agents [17] that lead to redox imbalance, the disruption of the flow of the electron transport chain, and oxidative stress in plants [18]. Plant production of ROS intensifies not only under water stress [19] but also upon exposure to abiotic stresses such as high temperatures [20], heavy metals [21], high salt concentrations [22], and herbicide application [23,24].

Plants have developed defense systems to mitigate damage from ROS, including enzymes such as superoxide dismutase (SOD), catalase (CAT), peroxidase (POD), ascorbate peroxidase (APX), and glutathione reductase (GR) [25]. In addition to these endogenous systems, the foliar application of low concentrations of organic compounds such as amino acids can minimize the oxidative damage caused by water stress. Amino acids are constituents of proteins and precursors of several regulators of plant metabolism [26]. According to [27], plant production of amino acids is reduced under stress conditions. Nutritional supplements that contain amino acids, such as proline, glycine, arginine [28], glutamate [29], and alanine, can improve plant development under biotic and abiotic stresses [18], such as root asphyxia, drought, and extreme temperatures [30,31], and minimize increases in energy expenditure [32]. 

Mineral nutrition also plays an important role in plant stress tolerance [33,34,35]. Nutrients such as potassium (K), magnesium (Mg), and phosphorus (P) can increase SOD, CAT, POD, and APX activity in plants and play important roles in cellular activities such as stomatal control, osmotic adjustment, enzyme activation, and amino acid synthesis [36,37]. The macronutrient requirements of crops are typically met by applying fertilizers to the soil because roots have high nutrient absorption. However, foliar fertilization can be a more effective and viable form of supplementation for plants [32]. This is a practice that could increase the efficiency of nutrient use, as the nutritional solution is applied directly to the plant, not to the soil, and at specific moments of demand by the crop, which generates greater efficiency in the use of the nutrients by the plants [38,39].

Although the roles of amino acids and mineral nutrients in the production of essential plant metabolites are well known, the specific mechanisms by which these molecules mitigate stress in plants are unclear [40,41]. A better understanding of the metabolic and structural functions of organic nutrients and the plant’s requirements for these compounds during different phenological stages would facilitate improvements in crop management, quality, and productivity [42,43]. 

This study evaluated the impact of the foliar application of an abiotic stress protection complement (ASPC) on sugarcane productivity under water stress. It was hypothesized that ASPC application would positively influence antioxidant system activity, biometric and technological parameters, biomass, and sugarcane productivity.

## 2. Results

### 2.1. ASPC Application Increases Antioxidant Enzyme Activity and Reduces Oxidative Stress

The activities of the antioxidant enzymes SOD, POD, and CAT were significantly (*p* ≤ 0.10) higher in the ASPC treatment than in the control at sites 1 and 2 (Figure 1A,B,D). ASPC application increased SOD activity by 24.04% and 78.13% and POD activity by 54.41% and 18.39% compared with the control, respectively, corresponding to 19.5 and 9.3 units g^−1^ protein for site 1 and 2.72 and 2.61 μmol min^−1^ g^−1^ protein for site 2 (Figure 1A,B). Conversely, ASPC application reduced the activity of PPO, an oxidizing enzyme that causes samples to darken in the presence of H_2_O_2_, at site 1. PPO activity decreased by 58% compared with the control (646.5 μmol catechol min^−1^ g^−1^ protein) (Figure 1C). ASPC application increased CAT activity by two times when compared with the control (0.28 μmol H_2_O_2_ min^−1^ g^−1^ protein) at site 1 (Figure 1D).

ASPC application increased Trolox-equivalent antioxidant capacity (DPPH), which measures the antioxidant capacity of a given plant tissue sample. Compared with the controls (9.82 and 3.38 mg TE g^−1^), ASPC application significantly (*p* ≤ 0.10) increased by 82.3% at site 1 and 83.7% at site 2 (Figure 1E). The concentration of MDA, a marker of oxidative stress, was lower in the ASPC treatment. Compared with the controls (24.9 and 33.14 nMg^−1^ protein), ASPC application significantly (*p* ≤ 0.10) decreased malondialdehyde (MDA) by 23% at site 1 and 1.65% at site 2 (Figure 2A). In line with the increases in SOD, CAT, and POD activity, ASPC application reduced H_2_O_2_ levels significantly (*p* ≤ 0.10), compared with the controls at 24.32 and 23.71 µmol g^−1^ FW. The rate of H_2_O_2_ decreased by 14.4% and 52.4% at sites 1 and 2, respectively (Figure 2B).

### 2.2. ASPC Application Increases Sugarcane Leaf Metabolites

The leaf reducing sugar of sugarcane was significantly higher (*p* ≤ 0.10) in the ASPC treatment than in the control (Figure 3). At sites 1 and 2, leaf reducing sugars increased by an average of 8.4% for site 1 and by two times for site 2 in the ASPC treatment compared with the controls (2.14 and 0.97%), respectively. Total soluble sugars were higher in the ASPC treatment b average of 10% for both sites 1 and 2 and 44.6% for site 3 compared to the control (0.70%, 1.37%, and 0.92%, respectively) (Figure 3A,B). In addition, leaf starch content was higher in all sites for control treatments. The higher rate was 2.5% (site 2), decreasing to 12.5% for ASPC treatment at this site (Figure 3C). In contrast to the results for starch, leaf sucrose increased in all sites in the ASPC treatment. Its higher rate was observed at site 3, increasing by 45% when compared with the control (0.64%) (Figure 3D).

### 2.3. ASPC Application Improves Sugarcane Stalk Technological Parameters

At sites 1, 2, and 3, ASPC application significantly (*p* ≤ 0.10) increased the sucrose content (%) in the stalk by 10%, 1.2%, and 2.9%, respectively, compared with the control, which had sucrose contents of 13.8%, 15.27%, and 16.23%, respectively (Figure 4A). 

In contrast to the results for sucrose concentration and purity, fiber content decreased by 3.9% in the ASPC treatment compared with the control (control value of 12.95%) at site 3. At site 2, fiber content increased by 2.6% in the ASPC treatment compared with the control (control value of 13.62%) (Figure 4C).

Sugarcane purity (%) was higher in the ASPC treatment at all sites (Figure 4B), with increases of 9.4%, 0.5%, and 0.8% compared with the control at sites 1, 2, and 3, respectively. The purity (%) control values were 85.65%, 81.02%, and 88.19%, respectively. Thus, ASPC application improved the quality of the raw material for industrial use, with average juice purity values above 80% at sites 1, 2, and 3.

Due to the increase in sucrose content, the content of reducing sugars decreased in the ASPC treatment at all sites (Figure 4D). The higher reducing sugar rate is presented at site 1, decreasing by 44% compared to the control at 0.52%.

The sucrose and reducing sugar contents were used to calculate TRS. Since ASPC application significantly (*p* ≤ 0.10) increased the sucrose content at sites 1, 2, and 3, TRS increased by 9%, 1.08%, and 2.5%, respectively, compared with the control (136.07, 151.8 and 159.2 kg Mg^−1^, respectively) (Figure 4E).

### 2.4. ASPC Application Increases Sugarcane Biometric Parameters

Among the biometric parameters, ASPC positively affected (*p* ≤ 0.10) stalk height (Figure 5A) at all sites, with average gains of 5% at site 1, 2.15% at site 2, and 3.7% at site 3 compared with the control (control values of 2.27, 2.28, and 2.33 m, respectively). In addition, ASPC application increased stalk diameter by 4.3% and 4.2% compared with the control (25.62 and 27.25 mm) at sites 1 and 3, respectively (Figure 5B).

### 2.5. ASPC Application Increases Stalk and Sugar Yields 

ASPC application significantly increased stalk yield, with average gains of 12.95 Mg ha^−1^ at site 1, 3.25 Mg ha^−1^ at site 2, and 11.54 Mg ha^−1^ at site 3 compared with the corresponding controls (control values of 73.27, 83.93, and 90.51 Mg ha^−1^), respectively (Figure 6A). Because sugar yield is calculated from the product of TRS and stalk yield, increases in these two parameters increase sugar yield. Sugar yield increased by averages of 28.5%, 4.78%, and 13.56% at sites 1, 2, and 3, respectively, compared with the control (control values of 9.97, 12.74, and 14.41 Mg ha^−1^) (Figure 6B).

### 2.6. ASPC Application Increases Biomass and Energy Production

In general, ASPC application increased biomass production (bagasse and trash) at all sites (Figure 7). Bagasse and trash production had the highest gains in the ASPC treatment at site 1, with gains of 1.62 Mg ha^−1^ and 1.08 Mg ha^−1^ compared with the control (9.36 and 6.17 Mg ha^−1^), respectively (Figure 7A,B). Consistent with the increase in biomass production, ASPC application increased energy production at all sites, with average gains of 15.3%, 5.5%, and 10.1% at sites 1, 2, and 3, respectively, compared with the control (76.8, 91.5, and 95.6 MWh) (Figure 7C).

### 2.7. Principal Component Analysis (PCA) among Sugarcane Parameters

Principal component analysis (PCA) identified the most important parameters that explain the variations in stalk and sugar yields. Since the metabolic and enzymatic attributes followed similar patterns at the different locations, the sites were grouped for analysis. The PCA considered the attributes leaf rates of reducing sugars, starch, sucrose, and total sugars and the activity of SOD, POD, PPO, CAT, MDA, Trolox, and H_2_O_2_ (Figure 8a). The correlations ≥ |0.70| were positive for leaf sucrose rates and the activities of SOD, POD, CAT, and Trolox. By contrast, the negative correlations ≥ |0.70| were attributed to the rates of starch, MDA, and H_2_O_2_ in the leaves (Table 1).

Figure 8b shows the distribution of the sites with (ASPC; green) and without (control; red) the application of an abiotic stress protection complement. The samples grouped on the left side were influenced by the rates of starch, MDA, and H_2_O_2_ in the leaves; therefore, they represented control treatments. Conversely, the samples grouped on the right side were influenced by higher leaf sucrose rates and the activities of SOD, POD, CAT, and Trolox; therefore, they represented the treatments under ASPC. Considering the average of the three locations, the treatments under ASPC were associated with stalks and sugar yield parameters.

## 3. Discussion

Water deficiency directly affects plant metabolism and physiology, including gas exchange and water and nutrient absorption, resulting in low growth rates, reduced photosynthesis and photoassimilate production, and lower plant biomass [16,28,44,45,46]. In this study, we evaluated the ability of the foliar application of ASPC, which contains amino acid-based organic compounds as the main ingredient but also contains macronutrients in order to mitigate water stress in sugarcane. 

In the three years of the experiments at sites 1, 2, and 3, ASPC was applied during a period when the climatic water balance indicated water scarcity (Figure 9). This period of water stress was most intense between June and July when precipitation was 25.4 mm at site 1, 10 mm at site 2, and 15.9 mm at site 3, indicating rainfall deficiency. The application of ASPC favorably influenced all parameters evaluated, including at site 2, where the sugarcane plants were the fifth ratoon. 

The combination of amino acids and nutrients in ASPC might have specific benefits that favor the alleviation of stress. According to [47,48], amino acids act as chelators to optimize the absorption of nutrients that play key roles in photosynthetic stability, especially Mg, P, nitrogen (N), and K. Photosynthetic cells are the main site of ROS production in plants [49], and stabilizing the photosynthetic apparatus is crucial to reducing damage caused by lipid peroxidation by ROS. This stabilization is mainly related to ROS scavenging by antioxidant enzymes (SOD, POD, and CAT) (Figure 1A,B,D) but may also be linked to the functions of the nutrients present in ASPC in primary metabolism. Mg is a central element of chlorophyll and thus is essential for chlorophyll synthesis [50]. Mg also activates enzymes in primary metabolism that are related to photosynthetic activity and directly influence plant growth and development, such as glutathione synthase, ribulose 1,5-biphosphate (RUBISCO), phosphoenolpyruvate carboxylase/oxygenase (PEPcase), RNA polymerase, protein kinases, phosphatases, and ATPases [51,52]. K is important for stomatal opening and closure and in the osmoregulation of cell content [53,54,55]. K also promotes photosynthesis indirectly through its effects on the architecture and functioning of the ribosome, an important organelle for protein synthesis [56,57]. Inorganic P (Pi) is important for photosynthesis because it serves as a phosphate transporter from the chloroplast inner membrane during the release of triosephosphate molecules (glyceraldehyde 3-phosphate and dihydroxyacetone phosphate) to the cytosol [38,39]. Finally, N is a component of chlorophylls, nucleic acids, amino acids, proteins, plant hormones, coenzymes, and secondary metabolites, which evidence the importance of this macronutrient in our findings and in other studies [25,43].

The efficiency of water use by plants is often directly linked to the proper functioning of the enzymatic antioxidant system because oxidative stress and increased ROS accumulation are a direct response to water stress [58,59]. The foliar application of ASPC increased the activity of antioxidant enzymes (CAT, POD, and SOD; Figure 1) [60,61,62]. Increases in SOD activity are associated with reduced ROS levels [63,64]. SOD dismutates O_2_^−^• to O_2_ and H_2_O_2_ and is the first line of defense against ROS in plant cells. POD and CAT convert the H_2_O_2_ generated by SOD into H_2_O, decreasing oxidative stress [65,66,67]. Conversely, PPO activity decreased in plants in the ASPC treatment compared to the control (Figure 1C). PPO catalyzes the formation of quinones and acts on phenolic compounds in the presence of oxygen, resulting in the oxidation and hydroxylation of phenols to produce melanoidins, brown pigments that cause tissue darkening [68]. According to [69], PPO is present in plastids and the cytoplasm under tissue degeneration conditions. The higher PPO activity in control plants implies greater levels of oxidative damage and lipid peroxidation in cells (Figure 2A,B).

The evaluation of MDA and H_2_O_2_ levels (Figure 2A,B) showed that the nutrients in ASPC not only aided primary metabolism but also activated the antioxidant system and reduced oxidative stress, ensuring a better response to water stress. Total antioxidant capacity as measured by Trolox DPPH was higher in the ASPC treatment than in the control (Figure 1E), indicating that ASPC activated the antioxidant system to eliminate ROS in leaves [63,64,70] and increase cellular phenol levels [71,72,73]. ASPC application reduced the level of MDA, the main product of lipid peroxidation, in response to water deficit. This metabolite is formed by the action of ROS on cell membranes, which results in the oxidation of polyunsaturated fatty acids and causes irreversible cell damage [74,75]. As water stress increases, plants increase ROS levels and, consequently, lipid peroxidation. The effects of ROS are particularly severe in organelles such as chloroplasts and mitochondria, which have extensive oxidative metabolism [76]. The foliar application of ASPC likely promoted the metabolic activation of the plant’s antioxidant system, reducing oxidative damage from water stress.

ASPC application increased total soluble sugar content but decreased starch content in sugarcane leaves. Starch is not only a reserve carbohydrate but also has a dynamic role in metabolism [77,78]. As the ability of the plant to respond to water stress improves, ROS levels are better regulated and photosynthesis becomes more stable. This ensures that the plant has a greater distribution of intracellular photoassimilates, which will increase cellular osmoprotection [77,78]. Studies suggest that in plants with better stress responses, starch levels decrease, and the levels of sugars related to the osmoregulation of cell content, such as maltose (which is directed to proline synthesis), increase [77,79,80]. The decrease in starch content in the ASPC treatment (Figure 3C) may reflect the plant’s use of reserve carbohydrates in response to water stress. 

The increased consumption of starch reserves in leaves results in the release of glucose and other cell components, increasing levels of reducing sugars in source tissues [81]. Studies of other plant species have shown that applying amino acids increases the levels of carbohydrates and polysaccharides in stressed plants [47,81,82]. By regulating the balance between stomatal opening and closure, amino acids enhance carbon fixation and the production of substrates for the synthesis of new carbohydrates [78]. This reaction is essential for the maintenance of osmoprotective balance in the cell. Carbohydrate production decreases the free energy of water and prevents cells from losing water to the external environment [83].

Reserve carbohydrate mobilization in the cytosol produces monosaccharides that are used in sucrose synthesis by sucrose-phosphate synthase (SPS) [77,84]. This may explain the increase in intracellular sucrose levels in sugarcane leaves in the ASPC treatment. These monosaccharides are important substrates for protective proteins and the synthesis of other components that protect against oxidative stress [25]. Sugarcane plants that received foliar ASPC had high stalk sucrose content (Figure 4A). By optimizing the antioxidant system and water use efficiency and reducing interference from plant abiotic stress, ASPC application increased sugarcane juice quality, which is determined by sucrose content, purity, TRS, reducing sugar content, and fiber percentage [85]. Higher sucrose content in the stalks improves raw material quality, as increases in sucrose are associated with decreases in glucose and fructose (reducing sugars) and help boost juice purity above the minimum of 80% required by the industry for sugarcane milling [86,87]. Although fiber content is negatively correlated with juice purity, the increase in fiber content in the ASPC treatment (Figure 4C) did not reduce the sucrose concentration. Moreover, the fiber content in the ASPC treatment did not exceed levels considered appropriate in the industry, i.e., between 12% and 13% [87,88]. Collectively, the improvements in these parameters increased sugar production (TRS) in kg Mg^−1^. The positive effects of ASPC application on raw material quality also probably reflect the presence of macronutrients, especially Mg and K, which participate in sucrose metabolism and transport.

ASPC application also increased stalk height and diameter (Figure 5A,B). The growth rate typically decreases when water stress occurs during a critical period of sugarcane development [84,85]. The smaller stalk diameter in the control treatment can be explained by the greater competition for photoassimilates between the leaves and stalk of the plant. Amino acids directly impact plant nutrition by increasing the efficiency of nutrient absorption, transport, and assimilation [48,62,89]. The application of ASPC may have improved the water use efficiency of sugarcane, thereby increasing the photosynthetic rate and the transport of sugars that provide metabolic energy for stalk growth. Previous studies of the foliar application of nutrients and amino acids to C4 crops have also reported increases in plant height and diameter [90,91].

Stalk and sugar yields were highest in the ASPC treatment. The application of amino acids reduced plant energy expenditure throughout the season, increased antioxidant activity, and improved the efficiency of protection against ROS. The higher crop yield can be explained by the increases in amino acids, lipids, carbohydrates, and cellulose as a result of foliar application [92]. Therefore, the higher yield of sugarcane treated with ASPC is probably linked to the higher production of components required for plant growth, protein synthesis, and osmoregulation, resulting in higher water stress tolerance.

Bagasse and trash production are directly affected by variations in plant vegetative growth. The accumulation of dry mass in the stalk may be associated with a greater demand for carbon by the stalk, which is an important sink for photoassimilates [93]. The stalk and fiber yields were used to calculate bagasse at 50% moisture, while straw was calculated considering 140 kg of trash per Mg of stalk [94]. ASPC application clearly increased bagasse and trash production, improving the energy potential of sugarcane production, even under water-stress conditions.

The use of ASPC as a management practice to mitigate water stress in sugarcane increased productivity even under adverse environmental conditions, as supported by our PCA analysis. ASPC acts directly on plant physiology, mobilizes defense enzymes, creates adequate conditions for metabolic functions, and promotes the synthesis of essential carbohydrates for full crop growth and development.

## 4. Materials and Methods

### 4.1. Experimental Area

The experiments were performed at three different sites in the south-central region of Brazil during the driest period of the year, between June and August, in 2018, 2019, and 2020 (different sites each year). The experiments were carried out on sugarcane (*Saccharum* spp. hybrid) fourth ratoon (site 1, 2018), fifth ratoon (site 2, 2019), and third ratoon (site 3, 2020). At site 1, sugarcane variety RB855536, which has excellent ratoon regrowth, high tillering, high sucrose content, and medium-to-late ripening, was cultivated. At sites 2 and 3, sugarcane variety SP80-3280, which has high sucrose content and ratoon yield, moderate tillering, large ratoon regrowth, and medium-to-late ripening, was grown. Site 1 is located in the municipality of Ponta Porã (22°20′09′′ S, 55°06′51′′ W, average altitude 755 m) and belongs to the BP Bunge mill group. Site 2 is located in the municipality of Pradópolis (21°21′34′′ S, 48°03′56′′ W, average altitude 538 m), and site 3 is located in the municipality of Motuca (21°30′30′′ S, 48°09′12′′ W, average altitude 538 m); both sites belong to the São Martinho mill group. According to the Köppen–Geiger climate classification system [95], the climate is humid subtropical (Cfa) at site 1 and tropical savanna (Aw) at sites 2 and 3. The average annual temperature is 21.3, 23.4, and 21.6 °C at sites 1, 2, and 3, respectively, and the average precipitation is 1700, 1419, and 1344 mm (Figure 9). The soil is classified as Rhodic Eutrodaf at site 1, Rhodic Eutrudox at site 2, and Hapludox Rhodic at site 3 [96]. The chemical characteristics of the soil were determined before the installation of the field experiments. Ten soil subsamples were collected between the ratoon rows at each site and pooled to form a composite sample. The chemical characteristics of the soil in the experimental area are presented in Table 2.

### 4.2. Experimental Design and Description of the Treatments

The experimental design was a randomized complete block (RCB) with two treatments and twelve replicates, giving a total of 24 experimental plots. Each plot had 8 10 m-long rows with an inter-row spacing of 1.5 m. The treatments consisted of (1) control (no application of ASPC) and (2) ASPC (application of ASPC). At all sites, the treatments were applied in June, at the beginning of the maturation stage, and the sugarcane was harvested in October.

ASPC was applied at the full manufacturer-recommended dose of 6 L a.i. ha^−1^ in a water volume of 100 L ha^−1^. The concentrations of the components of ASPC are given as % (*p*/*p*) in Table 3 and Table 4. ASPC is classified as a mixed mineral foliar fertilizer and has an average density of 1.28 g/mL at 20 °C (Ubyfol Agroquimica S.A., Uberaba, Brazil). The applications followed the manufacturer’s specifications and were performed under adequate environmental conditions with the help of a pressurized tank (CO_2_) equipped with a single 1/4KLC-9 brass fieldjet coupled to a 2.6 m-long rod, which allowed simultaneous and homogeneous application to four rows of plants (application range of 7.5 m) at a pressure of 344 kPa for each 100 L ha^−1^.

There were no problems with pests, weeds, or diseases at the experimental sites. Thus, crop management was carried out according to the recommendation for each site following the sugarcane mill’s calendar for sugarcane cultivation practices.

### 4.3. Leaf Collection for Enzyme Sampling 

Leaves were collected for enzymatic analysis between 9 a.m. and 10 a.m. 90 days after the application of ASPC. To collect the samples, we followed the recommendation in the literature [97] to collect 10 + 1 leaves and use only the middle third to evaluate the activities of superoxide dismutase (SOD) (EC 1.15.1.1), catalase (CAT) (EC 1.11.1.6), peroxidase (POD) (EC 1.11.1.7), and polyphenol oxidase (PPO) (EC 1.10.3.1).

### 4.4. Metabolic Changes in Sugarcane Leaves

Twenty sugarcane +1 leaves were collected and dried in a forced-air oven at 65 °C for 72 h. The material was then ground in a Wiley mill inside a sieve with a mesh diameter of 1 mm. The total sugars, soluble sugars, starch, and sucrose in the samples were measured, and the results were expressed in (%) for each 100 g of sample [98,99].

### 4.5. Oxidative Stress and Antioxidant Enzymes 

To prepare the leaf extract, 500 mg of plant tissue was first ground in liquid nitrogen with 10% (volume:volume) polyvinylpolypyrrolidone (PVPP). The ground tissue was combined at a ratio of 1:3 with 100 mM potassium phosphate buffer (pH 7.5) containing K_2_HPO_4_ (dibasic potassium phosphate) and KH2PO4 (monobasic potassium phosphate), 3 mM DTT (dithiothreitol), and 1 mM EDTA (ethylenediaminetetraacetic acid) [100]. The mixture was centrifuged for 10 min at 5000 rpm, and the supernatant was used as the crude extract. Before enzymatic analysis, the total protein concentration in the crude extract was determined using the methodology described by [101]. Briefly, 100 µL of crude extract was combined with 5 mL of Bradford reagent, and the absorbance at 595 nm was determined after 15 min.

Lipid peroxidation was evaluated based on malondialdehyde (MDA) levels following a previously reported method [38]. MDA was reacted with thiobarbituric acid (TBA), and the absorbance of the resulting TBA-MDA adduct was measured in a spectrophotometer at 532 nm. The concentration of MDA was calculated from a standard curve of 1,1,3,3-tetramethoxypropane (TPE), and a coefficient of 155 mM L^−1^ was used The results were expressed in nanomoles (nmol) of MDA per gram of fresh weight (g^−1^ FW). H_2_O_2_ was determined according to a reference calibration curve and expressed in mol g^−1^ FW [102].

SOD activity was evaluated as described previously [103]. The enzymatic activity was determined photochemically using an assay system consisting of methionine (13 mM; Sigma-Aldrich, St. Louis, MO, USA), EDTA (100 nM), riboflavin (2 µM; Sigma-Aldrich), and NBT (75 µM; Sigma-Aldrich) in 50 mM KPi buffer. The results were expressed in units (U) of SOD g^−1^ protein. CAT activity was assessed as described previously [104]. The enzymatic activity was determined by monitoring the consumption of H_2_O_2_ (250 µM), and the results were expressed in nmol min^−1^ mg^−1^ protein. POD activity was evaluated according to [105]. The assays were performed in a buffered solution with 200 mM KPi (pH 6.7), 10 mM EDTA, and 1% PVP (40). The enzymatic activity of free POD was measured colorimetrically by the formation of the 4-aminoantipyrine phenol complex. The results were expressed in µmol H_2_O_2_ min^−1^ g^−1^ FW. Finally, PPO activity was evaluated according to the literature [106]. The five-milliliter assay mixture for polyphenoloxidase activity consisted of the same assay mixture as that of peroxidase. The absorbance of the purpurogallin formed was taken at 420 nm. The results were expressed in µmol of catechol min^−1^ g^−1^ protein. Relative antioxidant capacity was measured as Trolox-equivalent antioxidant capacity using 2,2-diphenyl-1-picrylhydrasil (DPPH) and expressed in milligrams of TE per gram of FW [107].

### 4.6. Sugarcane Biometric Evaluations

Biometric evaluations were performed prior to harvest at the phenological stage of maturation. Twenty sugarcane plants were randomly collected from each plot. The stalk height of each plant was measured as the distance from the ground to the auricular region of the +1 leaf and expressed in meters; the stalk diameter was evaluated with the aid of a digital caliper [108].

### 4.7. Sugarcane Quality Evaluations

The stalks used for the biometric evaluations were also used for technological evaluations. The stalks were cleaned, defoliated, and sent to the payment for cane by sucrose content (PCTS) laboratory of the mill to determine the following parameters: sucrose concentration according to [86], % fiber (water insoluble dry matter in sugarcane), % purity (percentage of sucrose in total solids content in sugarcane juice), total reducing sugars (TRS; all forms of reducing or inverted sugars in sugarcane), and reducing sugars (RS %; reducing substances in cane and sugar products calculated as invert sugar, predominantly hexoses).

### 4.8. Stalk and Sugar Yield

Prior to harvest, two rows were defined within the useful area of the plot, and the plants within a 4 m length of each of these rows were used to evaluate stalk mass. The stalks were weighed, and the value was extrapolated to obtain the stalk yield in Mg ha^−1^. Then, the sugar yield (Mg ha^−1^) was calculated by multiplying the stalk yield (Mg ha^−1^) by TRS and dividing by 100.
$$Sugar\;Yield=\frac{Stalk\;Yield\times TRS}{100}$$

### 4.9. Biomass and Energy Production

The fiber and stalk yields were used to calculate bagasse production at 50% moisture according to [94], and the trash yield was calculated considering 140 kg of crop residue per Mg of stalks and 60% collection from the soil surface. 

Energy production was calculated according to [94] by considering the primary energy (1 MWh = 3600.00 MJ) contained in the crop residue and the bagasse: 1 Mg of crop residue has 4.96 MWh of primary energy and 1 Mg of bagasse has 4.94 MWh of primary energy.

### 4.10. Principal Component Analysis (PCA) Data Analysis

Multivariate analysis in the form of principal component analysis (PCA) was performed by using the statistical software package from StatStatsoft Statistica 7.0 Software 2005 [109]. To determine the factors in PCA, the Kaiser rule was used, considering eigenvalues > 1 or explaining over 85% of the total variance [110]. Correlations > |0.70| were considered [111].

### 4.11. Statistical Analysis

For statistical analysis, the normality of all data was first assessed using the Shapiro–Wilk test, and homoscedasticity was checked with Levene’s test (at the 5% level of significance; W ≥ 0.90, *p* ≤ 0.05). Site and year were considered separately due to differences in sugarcane ratoons (ages): fourth ratoon at site 1 in 2018, fifth ratoon at site 2 in 2019, fifth ratoon at site 1 in 2020, and third ratoon at site 3 in 2020. PFC application was considered a fixed factor. The block variable was considered a random variable. The mean data were compared by a *t*-test (*p* ≤ 0.1) in Minitab 19 software.

## 5. Conclusions

This field study demonstrates that the foliar application of a product combining macro-nutrients with amino acids can effectively increase sugarcane’s activity of antioxidant enzymes, quality, and stalk production under water stress. Mitigating stress by applying ASPC improved sugarcane stalk and sugar yields by up to 17.6% and 28.5%, respectively; even in the low improvements, sugarcane stalk and sugar yields achieved at least 3.9% and 4.8% gains. All these benefits were by promoting physiological and enzymatic responses that increased carbon assimilation and plant metabolism in ASPC-treated sugarcane. ASPC enhanced the sugarcane Trolox-equivalent antioxidant capacity and, in turn, antioxidant enzyme activity levels. In addition to reducing energy consumption by the plant, ASPC application promoted the plant’s antioxidant system and, in turn, increased the synthesis of carbohydrates, the concentrations of reducing sugars and fiber, and juice purity.

## Figures and Tables

**Figure 1 plants-13-00461-f001:**
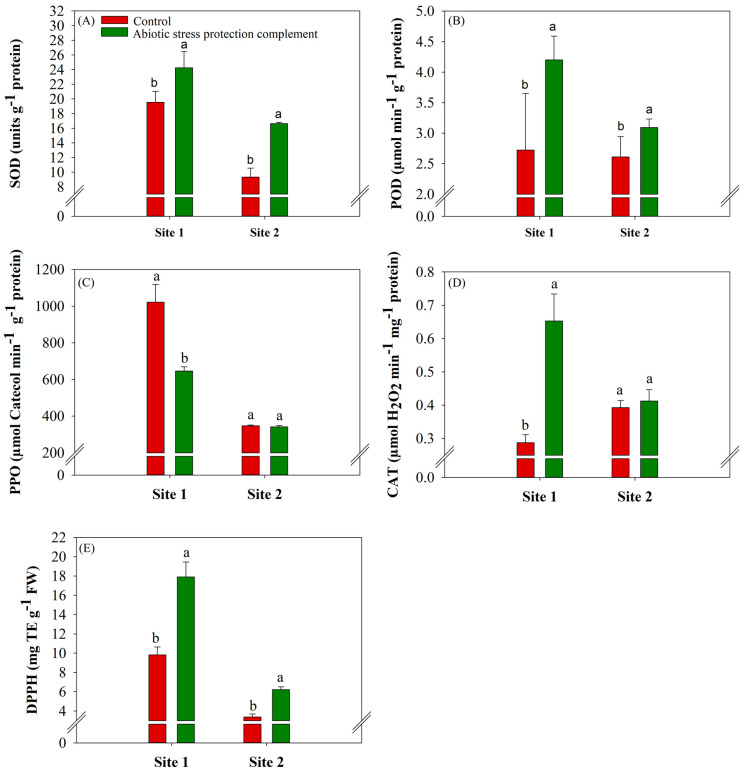
Effects of abiotic stress protection complement (ASPC) application on sugarcane antioxidant enzyme parameters at harvest: (**A**) superoxide dismutase (SOD), (**B**) peroxidase (POD), (**C**) polyphenol oxidase (PPO), (**D**) catalase (CAT) and (**E**) Trolox-equivalent antioxidant capacity (DPPH). The treatments were as follows: Control—no ASPC application; ASPC—application of ASPC in June, at the beginning of the drought season. Bars with the same letter do not differ by the ANOVA F-test (*p* ≤ 0.10).

**Figure 2 plants-13-00461-f002:**
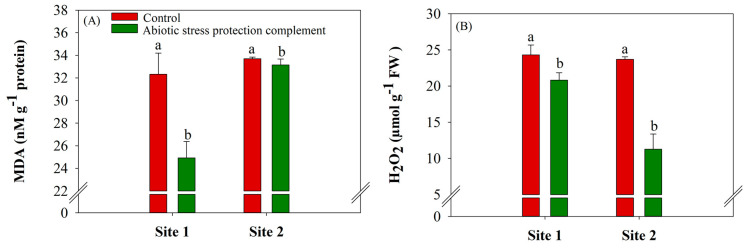
Effects of abiotic stress protection complement (ASPC) application on lipid peroxidation parameters at harvest: (**A**) malondialdehyde (MDA) and (**B**) hydrogen peroxide (H_2_O_2_) content. The treatments were as follows: Control—no ASPC application; ASPC—application of ASPC in June, at the beginning of the drought season. Bars with the same letter do not differ by the ANOVA F-test (*p* ≤ 0.10).

**Figure 3 plants-13-00461-f003:**
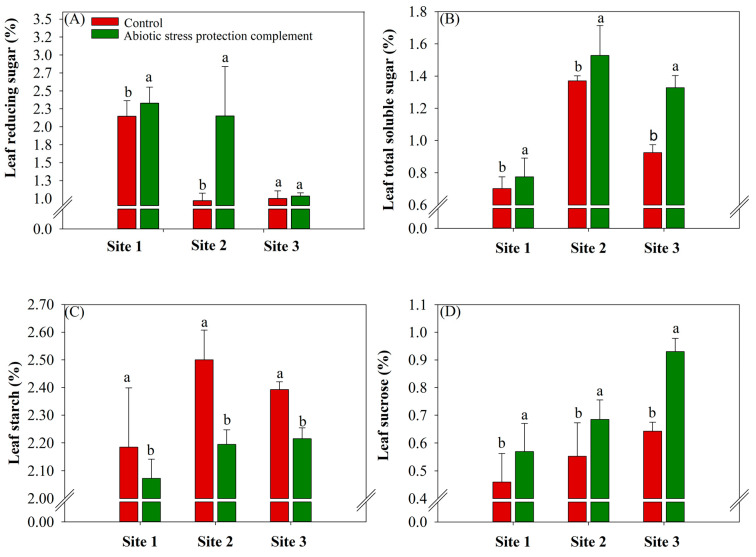
Effects of abiotic stress protection complement (ASPC) application on sugarcane leaf metabolic parameters at harvest: reducing sugars (%) (**A**), total soluble sugars (%) (**B**), starch (%) (**C**), and foliar sucrose (%) (**D**). The treatments were as follows: Control—no ASPC application; ASPC—application of ASPC in June, at the beginning of the drought season. Bars with the same letter do not differ by the ANOVA F-test (*p* ≤ 0.10).

**Figure 4 plants-13-00461-f004:**
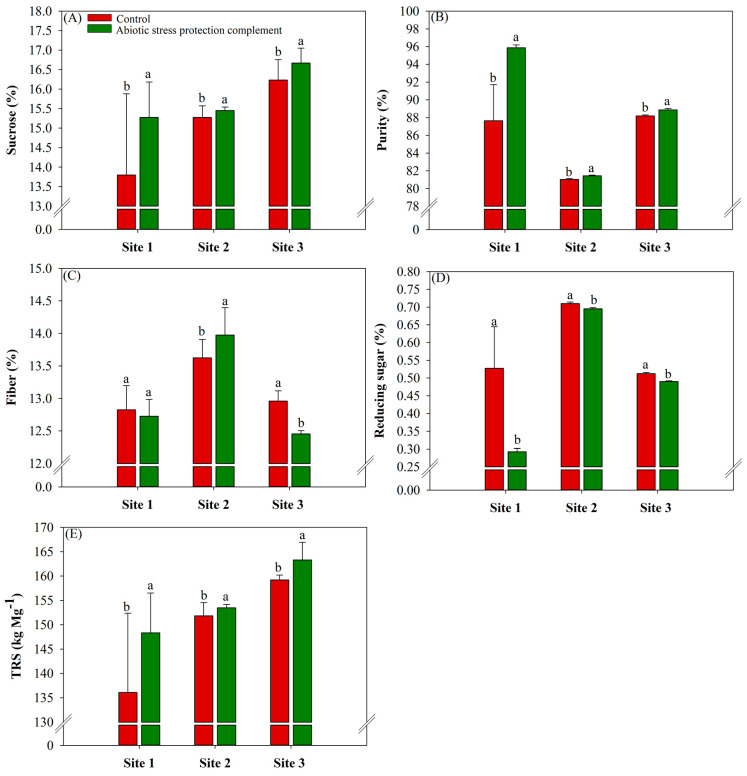
Effects of abiotic stress protection complement (ASPC) application on the technological parameters of sugarcane stalk at harvest: sucrose concentration (%) (**A**), purity (%) (**B**), fiber (%) (**C**), reducing sugars (%) (**D**), and total reducing sugars (kg Mg^−1^) (**E**). The treatments were as follows: Control—no ASPC application; ASPC—application of ASPC in June, at the beginning of the drought season. Bars with the same letter do not differ by the ANOVA F-test (*p* ≤ 0.10).

**Figure 5 plants-13-00461-f005:**
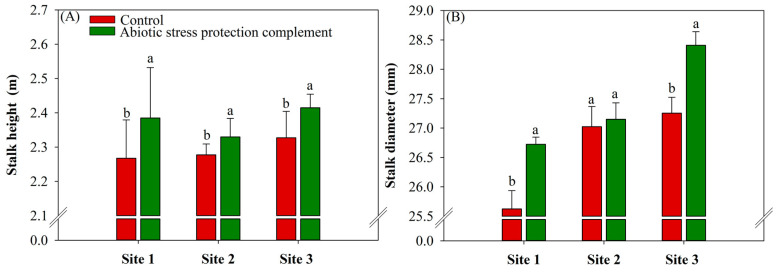
Effects of abiotic stress protection complement (ASPC) application on sugarcane biometric parameters at harvest: stalk height (m) (**A**) and stalk diameter (mm) (**B**). The treatments were as follows: Control—no ASPC application; ASPC—application of ASPC in June, at the beginning of the drought season. Bars with the same letter do not differ by the ANOVA F-test (*p* ≤ 0.10).

**Figure 6 plants-13-00461-f006:**
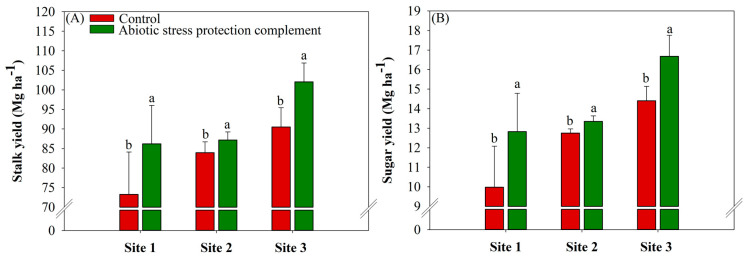
Effects of abiotic stress protection complement (ASPC) application on sugarcane yield parameters at harvest: stalk yield (Mg ha^−1^) (**A**) and sugar yield (Mg ha^−1^) (**B**). The treatments were as follows: Control—no ASPC application; ASPC—application of ASPC in June, at the beginning of the drought season. Bars with the same letter do not differ by the ANOVA F-test (*p* ≤ 0.10).

**Figure 7 plants-13-00461-f007:**
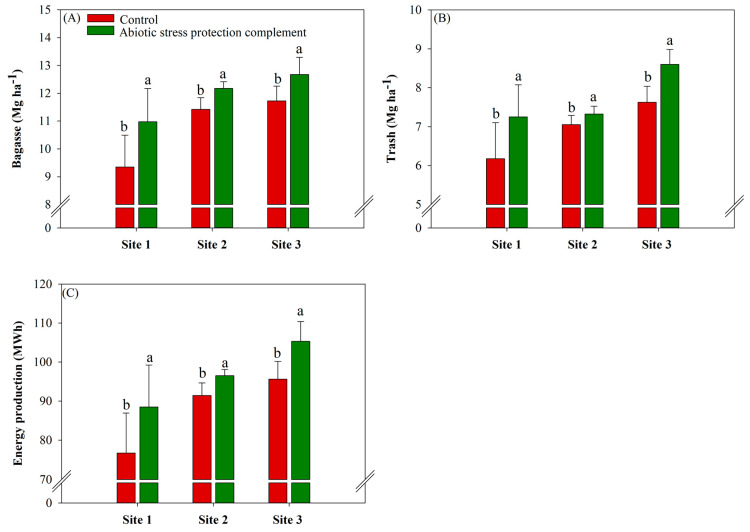
Effects of abiotic stress protection complement (ASPC) on sugarcane biomass and energy parameters at harvest: bagasse (Mg ha^−1^) (**A**), trash (Mg ha^−1^) (**B**), and energy production (MWh) (**C**). The treatments were as follows: Control—no ASPC application; ASPC—application of ASPC in June, at the beginning of the drought season. Bars with the same letter do not differ by the ANOVA F-test (*p* ≤ 0.10).

**Figure 8 plants-13-00461-f008:**
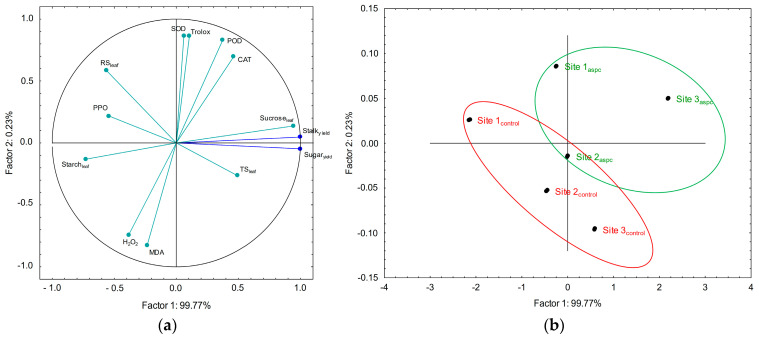
Projection of dataset based on correlations between leaf metabolic rates and enzymes (**a**) and sites (**b**) subjected to principal component analysis (PCA).

**Figure 9 plants-13-00461-f009:**
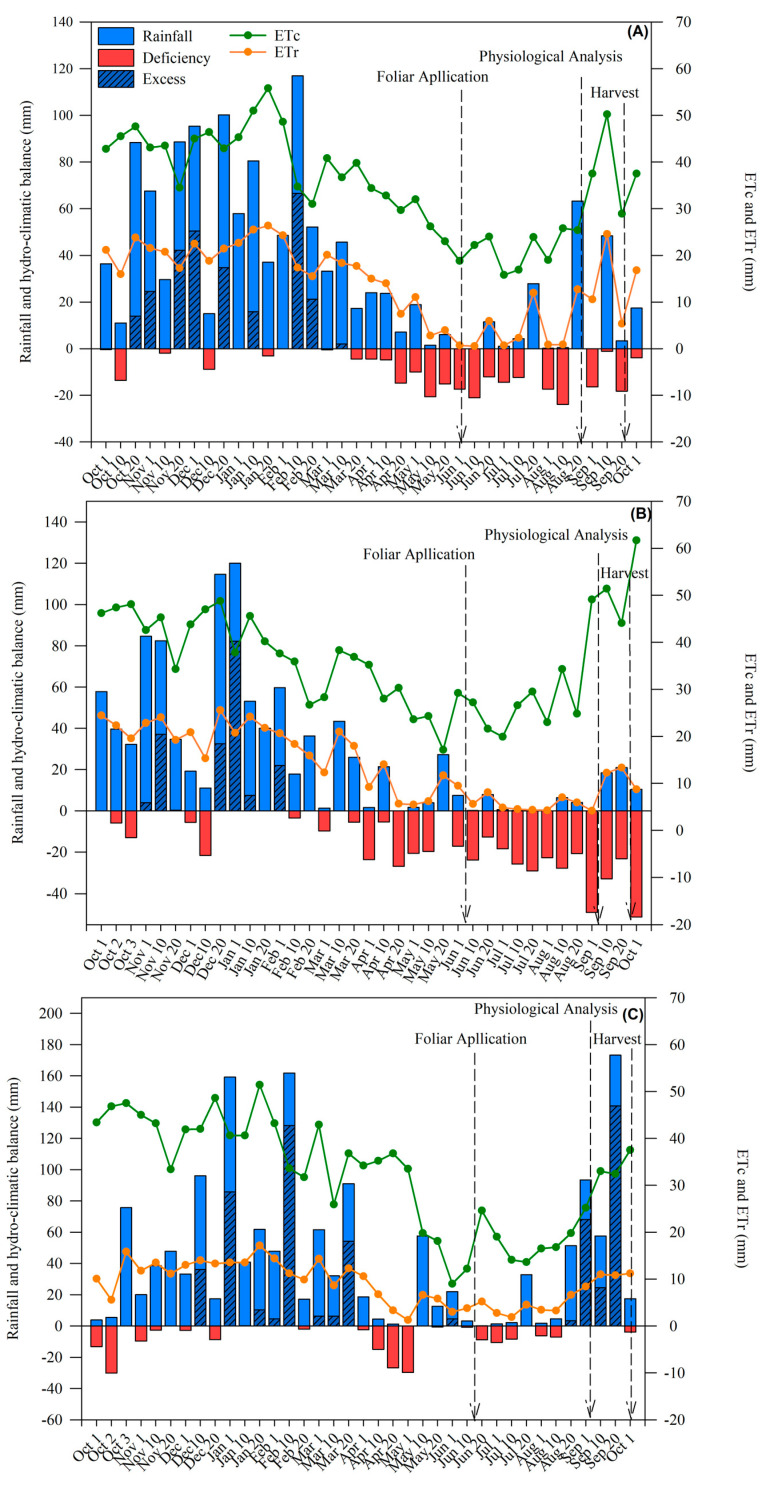
Climatological water balance at site 1 (**A**), site 2 (**B**), and site 3 (**C**) during the sugarcane crop cycles in 2018, 2019, and 2020, respectively. ETc, crop evapotranspiration; ETr, real evapotranspiration. The arrows indicate the timing of management operations and sampling.

**Table 1 plants-13-00461-t001:** Principal component analysis describing the influence of leaf metabolic rates and enzymes with sugarcane stalks and sugar yield at the average of three sites.

	Eigenvalue	Total Variance %	Cumulative Eigenvalues	Cumulative %
1	1.995	99.773	1.995	99.773
2	0.004	0.226	2.000	100.000
	Eigenvectors		Correlations	
Active ^2^	Factor 1	Factor 2	Factor 1	Factor 2
Stalk yield	0.707	0.707	0.998	0.047
Sugar yield	0.707	−0.707	0.998	−0.047
Correlations ^1^				
Average of three sites				
Supplementary ^3^	Factor 1	Factor 2		
RS_leaf_	−0.56	0.58		
Starch_leaf_	−0.72	−0.13		
Sucrose_leaf_	0.94	0.13		
TSl_eaf_	0.49	−0.26		
SOD	0.06	0.86		
POD	0.37	0.83		
PPO	−0.54	0.21		
CAT	0.46	0.70		
MDA	−0.23	−0.82		
Trolox	0.10	0.86		
H_2_O_2_	−0.38	−0.74		

^1^ Correlations ≥|0.70| are significant (Manly. 1994). ^2^ Actives: stalk yield and sugar yield. ^3^ Supplementary: leaf rates of reducing sugars (RS_leaf_), starch (Starch_leaf_), sucrose (Sucrose_leaf_), total sugars (TS_leaf_), and leaf enzyme activities of superoxide dismutase (SOD), peroxidase (POD), polyphenol oxidase (POD), catalase (CAT), malondialdehyde (MDA), Trolox, and hydrogen peroxidase (H_2_O_2_).

**Table 2 plants-13-00461-t002:** Soil classification and chemical characteristics of each experimental site.

Location	Site	Classification	Depth	pH	SOM	P_(resina)_	S	Al^+3^	H+Al^+3^	K	Ca	Mg	CEC	BS
				CaCl_2_	g dm^−3^	mg dm^−3^		mmol_c_ dm^−3^						%
Ponta Porã (MS) 2018	1	Rhodic Eutrodaf	0.00–0.20	5.1	34	13	18	1	36	0.77	42	11	90	58
			0.20–0.40	4.9	29	10	30	5	44	0.82	25	7	77	39
Pradópolis (SP) 2019	2	Rhodic Eutrudox	0.00–0.20	5.1	26	19	9	0	40	2.6	42	15	100	60
			0.20–0.40	5.1	23	16	29	0	42	1.7	33	15	91	53
Motuca (SP) 2020	3	Hapludox Rhodic	0.00–0.20	5.3	17	55	27	0	21	2.6	27	11	61	67
			0.20–0.40	5.0	14	34	33	1	25	0.8	17	6	49	49

Calculated from the equation: CEC (mmo_c_ kg^−1^) = Ca^2+^ + Mg^2+^ + K^+^ + (H+Al); Abbreviations: CEC, cation exchange capacity; m%, aluminum saturation; SOM, organic matter; V%, base saturation.

**Table 3 plants-13-00461-t003:** Concentrations of macronutrients in the abiotic stress protection complement (ASPC).

	Macronutrients					
	Mg	K_2_O	P_2_O_5_	N	S	TOC
Dose 6 L ha^−1^ (g)	153.6	460.8	460.8	76.8	245.76	122.8
Concentration (%)	2.0	6.0	6.0	1.0	3.2	1.6

TOC: total organic carbon.

**Table 4 plants-13-00461-t004:** Concentrations of amino acids present in the abiotic stress protection complement (ASPC).

	Amino Acids																	
	Gly	Glu	His	Ala	Tau	Pro	Asp	Lys	Arg	Leu	Ser	Thr	Tyr	Val	Phe	Met	Ile	Cys
Dose 6 L ha^−1^ (g)	180.5	139.8	135.2	113.7	104.5	87.6	86.8	69.9	68.4	57.6	44.5	40.7	37.6	37.6	29.9	26.9	23.8	10.8
Concentration (%)	2.35	1.82	1.76	1.48	1.36	1.14	1.13	0.91	0.89	0.75	0.58	0.53	0.49	0.49	0.39	0.35	0.31	0.14

Gly, glycine; Glu, glutamine; His, histidine; Ala, alanine; Tau, taurine; Pro, proline; Asp, asparagine; Lys, lysine; Arg, arginine; Leu, leucine; Ser, serine; Thr, threonine; Tyr, tyrosine; Val, valine; Phe, phenylalanine; Met, methionine; Ile, isoleucine; Cys, cysteine.

## Data Availability

Data are contained within the article.

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
