# Peer review of "Foliar Application of Amino Acids and Nutrients as a Tool to Mitigate Water Stress and Stabilize Sugarcane Yield and Bioenergy Generation"

_plants, 2024, doi:10.3390/plants13030461_

Round 1

Reviewer 1 Report

Comments and Suggestions for Authors

The manuscript presented for evaluation is a valuable resource for improving sugarcane yield height and quality. The introduction and literature references throughout the manuscript are appropriate and the information provided captures the essence of the research conducted. Likewise, the discussion.

However, I have a few comments on the remaining subsections:

Methodology:

I suggest placing Tables S1, S2 and S3 in the text of the manuscript where the authors cite them. The paper will not get significantly longer and the reception of the manuscript will be better.

The methodology should also strongly ring out that the study was conducted over 3 years but in a different location each year. The description itself suggests 3 years in 3 locations.

The description of the weather is prepared in detail, but I am wondering how the values presented relate to multi-year weather values?

Calculation of sugar yield please provide as a formula.

Subsections 4.8 and 4.9 have identical wording.

No methodology for biomass and related parameters (to the results described in subsection 2.6).

The methodology for DPPH is indicated as TEAC and the graph is signed as DPPH, alignment is needed.

Please cite the methodology for H2O2 measurement.

What guided the authors in setting p=0.1? The usual value is p=0.05.

The experiment includes plots sprayed with one, same dose plus a control. Judging by the description, this is an informal set-up of the experiment called 'only after with control group' two groups or areas (test and control) are selected and only the test area is affected. The explanatory variable is then measured in both areas simultaneously. The impact of the interaction is assessed by subtracting the value of the explanatory variable in the control area from its value in the test area. The basic assumption of such an experiment is that the two areas are identical in their behaviour towards the phenomenon under consideration. If this assumption is not true, there is the possibility of external variability in the interaction effect. And here the t-test is appropriate.

The authors state that they planned the year as a random factor. In agricultural experiments, it is customary not to consider year as a variable. If this is considered to be an experiment set up as a randomised complete block (RCB), then years should be considered as a block. If it were a randomised experiment (without a block), they should then use a one-way ANOVA. Such an experimental design is used when the experimental areas are homogeneous and differences due to uncontrolled factors are treated as random. I am leaning towards this arrangement.

Please comment in detail on how the experiment was planned and its statistical analysis.

Results: Sub-item 2.1: results and any information relating to box 3 are missing, why?

Conclusions: They are too general and do not reflect the dimension of the research carried out. They need to be enriched with more detailed conclusions of the research.

In summary, the very idea of the experiment is valid and scientifically sound. The research addresses an interesting and important issue. The results are interesting and well grounded in the reports of other authors. However, statistical analysis alone may change their tone. Apart from the subsection on the experimental design and the statistical analysis performed, the other comments do not affect the merits of the paper. However, without an explanation, I cannot indicate it for publication.

Author Response

Dear Plants reviewer, we thank you very much for taking your time to revise this manuscript. Please find the detailed responses below and the corresponding corrections in track changes in the revised files.

  1. I suggest placing Tables S1, S2 and S3 in the text of the manuscript where the authors cite them. The paper will not get significantly longer and the reception of the manuscript will be better.

Response: We have placed all the missing tables in the manuscript text.

  1. The methodology should also strongly ring out that the study was conducted over 3 years but in a different location each year. The description itself suggests 3 years in 3 locations.

Response: We have rewritten the sentence in the text for clarifying this statement.

  1. The description of the weather is prepared in detail, but I am wondering how the values presented relate to multi-year weather values?

Response: Data from location were not combined because sugarcane at sites (years) had different ages (cane ratoon). The sugarcane ages in the study was fourth ratoon (site 1, 2018), fifth ratoon (site 2, 2019) and third ratoon (site 3, 2020). Then, each year (site) has its specific climate condition and the sugarcane parameters values respond to that its related weather conditions. In addition, we can observe in hydric balance (Figure 1) that all years had pluviometry rates lower than the crop cycle requirements from 1800 to 2200 mm (i.e., 1768.4 mm; 1228 mm and 867,6 mm, respectively sites 1, 2 and 3), as well as the rainfall deficiency registered in the days following the ASPC application. Thus, all sugarcane parameters are described under drought condition in different years. We also have added information about the climate condition in the experimental regions in the introduction (lines 41-50) and in experimental area description in the material and methods (lines 370-390) to relate the sugarcane development in the study. 

  1. Calculation of sugar yield please provide as a formula.

Response: We have included it as a formula in line 492.

  1. Subsections 4.8 and 4.9 have identical wording.

No methodology for biomass and related parameters (to the results described in subsection 2.6).

Response: We are sorry for this mistake. We have included the correct methodology in the subsection 4.9.

  1. The methodology for DPPH is indicated as TEAC and the graph is signed as DPPH, alignment is needed.

Response: We have made the needed changes in line 22-115 and 468.

  1. Please cite the methodology for H2O2

Response: Done.

  1. What guided the authors in setting p=0.1? The usual value is p=0.05.

Response: The p-value of 0.1 was used mainly because it is a field experiment exposed to a large number of environmental variations, and, as it is an experiment with sugarcane crop, the sugarcane parameters have magnitudes with low variation. As a semi perennial crop, sugarcane remains for a long time in the field, which making crop susceptible to several adverse environmental factors. Thus it is coherent to use p-value of 0.1, and justified by the existence of the repeatability of two locations. Other studies also performed the statistical analysis using the same guide in agronomic field experiments on the same topic (p-value of 0.1), please see them below.

https://link.springer.com/article/10.1007/s12355-022-01177-5

https://www.sciencedirect.com/science/article/pii/S0167198722001118

  1. The experiment includes plots sprayed with one, same dose plus a control. Judging by the description, this is an informal set-up of the experiment called 'only after with control group' two groups or areas (test and control) are selected and only the test area is affected. The explanatory variable is then measured in both areas simultaneously. The impact of the interaction is assessed by subtracting the value of the explanatory variable in the control area from its value in the test area. The basic assumption of such an experiment is that the two areas are identical in their behaviour towards the phenomenon under consideration. If this assumption is not true, there is the possibility of external variability in the interaction effect. And here the t-test is appropriate.

The authors state that they planned the year as a random factor. In agricultural experiments, it is customary not to consider year as a variable. If this is considered to be an experiment set up as a randomised complete block (RCB), then years should be considered as a block. If it were a randomised experiment (without a block), they should then use a one-way ANOVA. Such an experimental design is used when the experimental areas are homogeneous and differences due to uncontrolled factors are treated as random. I am leaning towards this arrangement.

Please comment in detail on how the experiment was planned and its statistical analysis.

Response: We are sorry for the misunderstanding of the statistical analysis. We have revised this section and included error bars. We have explained how the statistical analyses were performed in the M&M.

All data were submitted to normality (Shapiro–Wilk) and homoscedasticity (Levene’s) tests (at the 5% level of significance; W ≥ 0.90, p < 0.05). Site and year were considered separately due to differences in sugarcane ratoons (ages): fourth ratoon at site 1 in 2018, fifth ratoon at site 2 in 2019, fifth ratoon at site 1 in 2020, and third ratoon at site 3 in 2020. PFC application was considered a fixed factor. The block variable was considered a random variable. The mean data were compared by a t-test (p ≤ 0.1) in Minitab 19 software.

Therefore, year was not considered as a random factor.

  1. Results: Sub-item 2.1: results and any information relating to box 3 are missing, why?

Response: Sorry, are you referring to site 3 when you ask for “box 3”?

If it is the case, we had technical problems with the equipment used to read the enzyme analyzes (UV Spectrophotometer), therefore, it would not be possible to carry out the enzymatic analyzes of sugar cane samples at the same phenological stage as in the years 2018 and 20219 (sites 1 and 2, respectively). Then, we have no data for site 3 in sub-item 2.1.

However, site 2 and site 3 are approximately 30 km away from each other, while site 1 is approximately 1000 km away from the other two sites. Therefore, we believe that the results from site 2 can represent site 3 well.

  1. Conclusions: They are too general and do not reflect the dimension of the research carried out. They need to be enriched with more detailed conclusions of the research.

Response: Thank you for your comment. We have improved the conclusions as suggested by reviewer.

Conclusion: “This field study demonstrates that foliar application of a product combining mac-ronutrients with amino acids can effectively increase sugarcane activity of antioxidant enzymes, quality, and stalk production under water stress. Mitigating stress by apply-ing ASPC improved up to 17,6% and 28,5% of sugarcane stalk and sugar yields, re-spectively; even in the low improvements, sugarcane stalk and sugar yields achieved at least at least 3,9% and 4,8% of gains. All these benefits were by promoting physio-logical and enzymatic responses that increased carbon assimilation and plant metabo-lism in sugarcane ASPC-treated. As it was the case of ASPC enhanced the sugarcane trolox-equivalent antioxidant capacity and, in turn, antioxidant enzyme activity level. In addition to reducing energy consumption by the plant, ASPC application promoted the plant's antioxidant system and, in turn, increased the synthesis of carbohydrates, the concentrations of reducing sugars and fiber, and juice purity.”.

Reviewer 2 Report

Comments and Suggestions for Authors

The ms plants-2788648 entitled Foliar application of amino acids and nutrients as a tool to mitigate water stress and stabilize sugarcane yield and bioenergy generation investigates an interesting topic, but the authors have to improve it before it can go for further process:

The authors should choose to use water stress or drought? Use only one term in your ms.

L25 productivity of what?

L24-26 Moreover, the application of ASPC increased productivity to 11.55 Mg ha-1 and sucrose accumulation to 2.26 kg Mg-1 , thus increasing the quality of the raw material. The authors here should compare it to the control treatment, so please revise.

L34       Drought results in changes in plant metabolism, morphology, and physiology [1–3]. Ref 2 and 3 are Spanish or something else, please replace ref 2 and 3 with the following relevant and recent citations as follows: “Drought stress impacts on plants and different approaches to alleviate its adverse effects” & “Additions of optimum water, spent mushroom compost and wood biochar to improve the growth performance of Althaea rosea in drought-prone coal-mined spoils”

L38-39 the authors can use one further citation “Sequential Application of Antioxidants Rectifies Ion Imbalance and Strengthens Antioxidant Systems in Salt-Stressed Cucumber”

L43-45 The authors here mentioned “under drought stress” and then mentioned again in same sentence “water stress [10],” What is the difference? Please add these citations for ROS and salinity “Glycine-betaine induced salinity tolerance in maize by regulating the physiological attributes, antioxidant defense system and ionic homeostasis” & “Activated Yeast Extract Enhances Growth, Anatomical Structure, and Productivity of Lupinus termis L. Plants under Actual Salinity Conditions”,  for heavy metals “Enhancing antioxidant defense system of mung bean with a salicylic acid exogenous application to mitigate cadmium toxicity”,

L52 Castro and Carvalho (2014), this is the wrong citation, follow the format of the journal for citing the references within the ms.

L57 xxxx, Authors are recommended to cite “Use of plant nutrients in improving abiotic stress tolerance in wheat”, “Interaction effects of nitrogen source and irrigation regime on tuber quality, yield, and water use efficiency of Solanum tuberosum L.”, “Biochar and its broad impacts in soil quality and fertility, nutrient leaching and crop productivity: A review”,

L74 under water stress, What kind of water stress? Can you please explain the treatment of the water stress here in a short sentence?

Figure 1 can be revised, particularly graph 1 in terms of the unit on the axes X

L95 (p < 0.10) is wrong, it should be (p 0.10) check this in whole ms please.

Can you please kindly add the standard error values about the columns in the Figures as error bars?

L146 and in some other places within the ms, the authors indicate (p < 0.10) for the significance differences, and this is wrong. If the p-value is 0.05 or lower, the result is trumpeted as significant, but if it is higher than 0.05, the result is non-significant. Thus, pleas e discuss this issue with someone who understands the satirical analysis and correct this issue in whole ms.

L194-196 Water deficiency directly affects plant metabolism and physiology, including gas exchange and water and nutrient absorption, resulting in low growth rates, reduced photo- synthesis and photoassimilate production, and lower plant biomass. Here I suggest authors to cite following “Foliar Applications of ZnO and SiO2 Nanoparticles Mitigate Water Deficit and Enhance Potato Yield and Quality Traits”, “Calibration and validation of AQUACROP and APSIM models to optimize wheat yield and water saving in arid regions”, “Morphological and Biochemical Response of Potatoes to Exogenous Application of ZnO and SiO2 Nanoparticles in a Water Deficit Environment”

L230-231 here the authors should make it clear for their results and other investigations to make it clear for the readers. Revise please.

L349 Do you think that The experimental design was a randomized complete block (RCB) with two treatment was correct?  

L410 wrong format for Fernandes (2003), follow the guidelines of the journal.

In Conclusions, please add some key values to highlight your results and Conclusions.

I suggest also, the author to write few sentences about the bioenergy using suitable references such as “Biomass yield and quality of bioenergy crops grown with synthetic and organic fertilizers”. “Feedstock quality and growth of bioenergy crops fertilized with sewage sludge”. “Towards sustainable intensification of feedstock production with nutrient cycling”

Good luck

Comments on the Quality of English Language

minor edits for English are needed

Author Response

Dear Plants reviewer, we thank you very much for taking your time to revise this manuscript. Please find the detailed responses below and the corresponding corrections in track changes in the revised files.

  1. The authors should choose to use water stress or drought? Use only one term in your ms.

Response: We have chosen the term water stress as it is more convenient for the reader undestanding, however, in some contexts we kept the term drought once we want to emphasize the weather condition, not the plant metabolic response caused by only water stress. 

  1. L25 productivity of what?

Response: Stalk yield. We have rewritten the abstract due to the allowed maximum word number in this section.

  1. L24-26 Moreover, the application of ASPC increased productivity to 11.55 Mg ha-1 and sucrose accumulation to 2.26 kg Mg-1, thus increasing the quality of the raw material. The authors here should compare it to the control treatment, so please revise.

Response: Thank you for your comment. We have revised it and changed the abstract according to the allowed maximum word number in this section.

  1. L34 Drought results in changes in plant metabolism, morphology, and physiology [1–3]. Ref 2 and 3 are Spanish or something else, please replace ref 2 and 3 with the following relevant and recent citations as follows: “Drought stress impacts on plants and different approaches to alleviate its adverse effects” & “Additions of optimum water, spent mushroom compost and wood biochar to improve the growth performance of Althaea rosea in drought-prone coal-mined spoils”

Response: We have done.

  1. L38-39 the authors can use one further citation “Sequential Application of Antioxidants Rectifies Ion Imbalance and Strengthens Antioxidant Systems in Salt-Stressed Cucumber”.

Response: We have done.

  1. L43-45 The authors here mentioned “under drought stress” and then mentioned again in same sentence “water stress [10],” What is the difference? Please add these citations for ROS and salinity “Glycine-betaine induced salinity tolerance in maize by regulating the physiological attributes, antioxidant defense system and ionic homeostasis” & “Activated Yeast Extract Enhances Growth, Anatomical Structure, and Productivity of Lupinus termis L. Plants under Actual Salinity Conditions”, for heavy metals “Enhancing antioxidant defense system of mung bean with a salicylic acid exogenous application to mitigate cadmium toxicity”,

Response: We apologize for the repetition error in terminologies, we have corrected this line. Drought is related to weather condition and water stress is related plant metabolic response caused by weather condition. 

  1. L52 Castro and Carvalho (2014), this is the wrong citation, follow the format of the journal for citing the references within the ms.

Response: We have done.

  1. L57 xxxx, Authors are recommended to cite “Use of plant nutrients in improving abiotic stress tolerance in wheat”, “Interaction effects of nitrogen source and irrigation regime on tuber quality, yield, and water use efficiency of Solanum tuberosum L.”, “Biochar and its broad impacts in soil quality and fertility, nutrient leaching and crop productivity: A review”,

Response: We have done.

  1. L74 under water stress, What kind of water stress? Can you please explain the treatment of the water stress here in a short sentence?

Response: The water stress treatment can be explained by the hydric balance (Figure 1), where we can see that all the years had pluviometry rates lower than the crop cycle requires within 1800-2200 mm (1768.4 mm; 1228 mm and 867,6 mm, respectively sites 1,2 and 3), besides of the rainfall deficiency (Figure 1) registered in the days following the ASPC application. Thus, all the parameters are described under drought condition (weather), in different years, however, with the same rainfall deficiency characteristic (Figure 1). Also, the levels of malondialdehyde (MDA) and hydrogen peroxide (H2O2) in control plants ensure the metabolic stress condition (water stress). We have added information about the climate condition in the experimental regions in the introduction (lines 41-50) and in sites description in the material and methods (lines 370-390). 

  1. Figure 1 can be revised, particularly graph 1 in terms of the unit on the axes X

Response: We have revised the units and added just the months referring to the crop cycle, not the entire full normal year from January to December. The dates are from 10 to 10 days for being able to register easily any anomaly in the rainfall regime.

  1. L95 (p < 0.10) is wrong, it should be (p ≤ 0.10) check this in whole ms please.

Response: We have fixed it.

  1. Can you please kindly add the standard error values about the columns in the Figures as error bars?

Response: Sure, we have added it.

  1. L146 and in some other places within the ms, the authors indicate (p < 0.10) for the significance differences, and this is wrong. If the p-value is 0.05 or lower, the result is trumpeted as significant, but if it is higher than 0.05, the result is non-significant. Thus, pleas e discusses this issue with someone who understands the satirical analysis and correct this issue in whole ms.

Response:  The p-value of 0.1 was used mainly because it is a field experiment exposed to a large number of environmental variations, and, as it is an experiment with sugarcane crop, the sugarcane parameters have magnitudes with low variation. As a semi perennial crop, sugarcane remains for a long time in the field, which making crop susceptible to several adverse environmental factors. Thus it is coherent to use p-value of 0.1, and justified by the existence of the repeatability of two locations. Other studies also performed the statistical analysis using the same guide in agronomic field experiments on the same topic (p-value of 0.1), please see them below.

https://link.springer.com/article/10.1007/s12355-022-01177-5

https://www.sciencedirect.com/science/article/pii/S0167198722001118

  1. L194-196 Water deficiency directly affects plant metabolism and physiology, including gas exchange and water and nutrient absorption, resulting in low growth rates, reduced photo- synthesis and photoassimilate production, and lower plant biomass. Here I suggest authors to cite following “Foliar Applications of ZnO and SiO2 Nanoparticles Mitigate Water Deficit and Enhance Potato Yield and Quality Traits”, “Calibration and validation of AQUACROP and APSIM models to optimize wheat yield and water saving in arid regions”, “Morphological and Biochemical Response of Potatoes to Exogenous Application of ZnO and SiO2 Nanoparticles in a Water Deficit Environment”

Response: We have done

  1. L230-231 here the authors should make it clear for their results and other investigations to make it clear for the readers. Revise please.

Response: We have clarified it: “Finally, N is a component of chlorophylls, nucleic acids, amino acids, proteins, plant hormones, coenzymes, and secondary metabolites, which evidence the importance of this macronutrient in our findings and other studies.”

  1. L349 Do you think that The experimental design was a randomized complete block (RCB) with two treatments was correct?

Response: Yes, because it is a field experiment exposed to a large number of environmental variations, and the two treatments were composed by 12 replicates

  1. L410 wrong format for Fernandes (2003), follow the guidelines of the journal.

Response: We have done.

  1. In Conclusions, please add some key values to highlight your results and Conclusions.

Response: We have improved the conclusion section.

  1. I suggest also, the author to write few sentences about the bioenergy using suitable references such as “Biomass yield and quality of bioenergy crops grown with synthetic and organic fertilizers”. “Feedstock quality and growth of bioenergy crops fertilized with sewage sludge”. “Towards sustainable intensification of feedstock production with nutrient cycling”

Response: We thank you for your suggestion.

Reviewer 3 Report

Comments and Suggestions for Authors

The manuscript deals with the evaluation of preparation composed of amino acids and nutrients in mitigation drought stress in sugarcane and impact on plant parameters. However, the study was carried out in field conditions where it is hard to stimulate drought stress. Precipitation between foliar application and physiological analysis was unstable, therefore, why Authors claim that plants were exposed to drought. In the Introduction justify the selection of sugarcane for this study. Highlight its importance in food production. Moreover, some references should be replaced by newer. Other comments are listed below:

L25: replace values by % differences between treatment and control

L45: generally pesticides are factors of abiotic stress. There are some studies referring to herbicides, fungicides and insecticides as the reason of abiotic stress. Replace references 12 and 13 by newer:

https://doi.org/10.3390/agronomy13051378

https://doi.org/10.1016/j.chemosphere.2022.136284

L78: why site 3 was omitted in this part?

L83-86: express units as g-1 protein

L114-115: determination of glucose and fructose was not described in Materials and Methods. Glucose and fructose are not the only reducing sugars in plants

L151: sugarcane stalk at harvest

L161: stalk diameter in Fig. 6 and in caption in L163

L179: correct the name of the heading

L182: production was the highest in site 3, as shown in Fig. 8

L327: Latin name in italics

L328-330: why two varieties were selected. It may influence the reliable comparison of the results

L345: add temperature in the Fig.1. Explain what Etc and ETr mean

L364: does it mean that no pesticides were applied?

L371: indicate full names of enzymes

L373-376 and L388-397: briefly describe the procedures

L422-426: it is the same description as in point 4.8

Author Response

Dear Plants reviewer, we Thank you very much for taking the time to review this manuscript. Please find the detailed responses below and the corresponding revisions/corrections highlighted/in track changes in the re-submitted files.

  1. The manuscript deals with the evaluation of preparation composed of amino acids and nutrients in mitigation drought stress in sugarcane and impact on plant parameters. However, the study was carried out in field conditions where it is hard to stimulate drought stress. Precipitation between foliar application and physiological analysis was unstable, therefore, why Authors claim that plants were exposed to drought.

Response: The water stress can be explained by the hydric balance (Figure 1), where we can see that all the years had pluviometry rates lower than the crop cycle requires within 1800-2200 mm (1768.4 mm; 1228 mm and 867,6 mm, respectively sites 1,2 and 3), besides of the rainfall deficiency (Figure 1) registered in the days following the ASPC application. Thus, all the parameters are described under drought condition (weather), in different years, however, with the same rainfall deficiency characteristic (Figure 1). Also, the levels of malondialdehyde (MDA) and hydrogen peroxide (H2O2) in control plants ensure the metabolic stress condition (water stress). We have added information about the climate condition in the experimental regions in the introduction (lines 41-50) and in sites description in the material and methods (lines 370-390).

  1. In the Introduction justify the selection of sugarcane for this study. Highlight its importance in food production.

Response: We have introduced the sugarcane justification in the introduction section.

  1. L25: replace values by % differences between treatment and control

Response: We have revised the abstract due to the maximum word permitted in this section.

  1. L45: Generally, pesticides are factors of abiotic stress. There are some studies referring to herbicides, fungicides and insecticides as the reason of abiotic stress. Replace references 12 and 13 by newer:

Response: We have done.

https://doi.org/10.3390/agronomy13051378

https://doi.org/10.1016/j.chemosphere.2022.136284

  1. L78: why site 3 was omitted in this part?

Response: We had technical problems with the equipment used to read the enzyme analyzes (UV Spectrophotometer), therefore, it would not be possible to carry out the enzymatic analyzes of sugar cane samples at the same phenological stage as in the years 2018 and 20219 (sites 1 and 2, respectively). Then, we have no data for site 3 in sub-item 2.1.

However, site 2 and site 3 are approximately 30 km away from each other, while site 1 is approximately 1000 km away from the other two sites. Therefore, we believe that the results from site 2 can represent site 3 well.

  1. L83-86: express units as g-1 protein

Response: We have fixed it, also in the figures 2 and 3.

  1. L114-115: determination of glucose and fructose was not described in Materials and Methods. Glucose and fructose are not the only reducing sugars in plants

Response: We have deleted it from the results in L133-134.

  1. L151: sugarcane stalk at harvest

Response: We have fixed it.

  1. L161: stalk diameter in Fig. 6 and in caption in L163

Response: We have fixed it.

  1. L179: correct the name of the heading

Response: We have fixed it.

  1. L182: production was the highest in site 3, as shown in Fig. 8

Response: The highest production was at site 3, however, the highest gains was at site 1. We have rewritten the sentence to make it clearer at 2.6 section.

  1. L327: Latin name in italics

Response: We have fixed it.

  1. L328-330: why two varieties were selected. It may influence the reliable comparison of the results

Response: The varieties were used according to the region and specific climatic conditions. Site 2 and site 3 are approximately 30 km away from each other, while site 1 is approximately 1000 km away from the other two sites. The latitudes are different, 1-degree difference, so sites 1 and 2 are above the Tropic of Capricorn, while site 1 is below this latitude. For this reason, varieties need to be specific to each region.

  1. L345: add temperature in the Fig.1. Explain what Etc and ETr mean

Response: We are sorry, but we were unable to add the temperature in figure 5 because the Y axis with the units of ETc (crop evapotranspiration) and ETr (real evapotranspiration) are not fully compatible with temperature. However, if it is convenient, we can create a specific graph just for temperature. We also added information to the figure legend, including the meaning of ETc and ETr.

  1. L364: does it mean that no pesticides were applied?

Response: We meant that the phytosanitary management followed the recommendations and needs of the crop, that is, as a preventive tool the use of pest management with biological control, especially to control Diatraea saccharalis and Mahanarva fimbriolata.

  1. L371: indicate full names of enzymes.

Response: We have done.

  1. L373-376 and L388-397: briefly describe the procedures

Response: We have done.

  1. L422-426: it is the same description as in point 4.8

We have fixed it.

Round 2

Reviewer 1 Report

Comments and Suggestions for Authors

I have no comments. The authors have comprehensively addressed all questions.

Author Response

Dear Plants reviewer, we thank you very much for taking your time to revise this manuscript.

Reviewer 2 Report

Comments and Suggestions for Authors

The authors improved their ms, and the ms can be accepted now.

Author Response

(The authors gave the same response as above.)

Reviewer 3 Report

Comments and Suggestions for Authors

The Authors have significantly improved the manuscript. I have only some comments:

check the abbreviation of molar concentration in point 4.5

L532: duplication of 'at least' 

Author Response

Dear Plants reviewer, we Thank you very much for taking the time to review this manuscript. Please find the detailed responses below and the corresponding revisions/corrections highlighted/in track changes in the re-submitted files.

  1. Check the abbreviation of molar concentration in point 4.5

Response: We have checked it.

  1. L532: duplication of 'at least'

Response: We have fixed it.
